# Research on Environmental Suitability Evaluation of the Transfer Spaces in Urban Subway Stations

**Zihan Wu [1], Xiang Ji [1,2,*], Xi Zhou [3] and Shuai Tong [1]**

1   School of Mechanics and Civil Engineering, China University of Mining and Technology,
    Xuzhou 221000, China
2   Jiangsu Collaborative Innovation Center for Building Energy Saving and Construction Technology,
    Jiangsu Vocational Institute of Architectural Technology, Xuzhou 221000, China
3   School of Civil Engineering and Architecture, Anhui University of Science and Technology,
    Huainan 232001, China
*   Correspondence: jixiang0615@yeah.net

**Abstract:** The transfer space realizes the connectivity of subway intersections. Passengers generally express that they have a poor experience in the use of this space, so improving the environmental suitability of transfer spaces at subway stations is a top priority. Based on a literature review and field research, this study established an environmental suitability evaluation system for transfer spaces and used the fuzzy comprehensive evaluation method to evaluate the environmental suitability of eight samples in Shanghai. The results showed that the evaluation results of the eight samples were ranked as follows: Hanzhong Road Station > People's Square Station > East Nanjing Road Station > Century Avenue Station > Xujiahui Station > Laoximen Station > Jiangsu Road Station > Shanghai Railway Station. Through the analysis of the relationship between the indicators, it was found that the environmental suitability of a transfer space is greatly affected by safety and convenience, while practicality, comfort, and aesthetics were found to have a weak influence on the suitability of transfer spaces. These evaluation methods and results provide a reference for the improvement of the environmental quality of subway transfer spaces in other cities.

**Keywords:** rail transit; station transfer; fuzzy comprehensive evaluation; environmental suitability

## 1. Introduction

Continued urbanization has resulted in an increasing urban population, leading to the rapid development of subways [1,2], which have gradually become one of the most common ways to commute [3,4]. The transfer space of a subway station assumes the function of the continuous transfer of passengers, and the closed characteristics of this space directly affect the physical and mental health, way-finding ability, and behavior choices in emergencies of passengers [5,6]. During holidays and rush hours, a large number of passengers gather, increasing waiting times and the danger of stampedes [7]. Therefore, it is urgent to improve the environmental suitability of the transfer spaces of subway stations and to effectively improve the travel experience of passengers in these transfer spaces.

Various research methods have been applied to the subway transfer space environment. Hernandez et al. [8] conducted field research on European subways and found that the internal environment of the subway station transfer space was an important factor affecting the transfer experience. Xu et al. [9] proposed a newly built mathematical model for streamlining transfer spaces and a generalized cost function evaluation method to quantify the choice of transfer routes within passenger stations. Li and Ge [10] applied the analytic hierarchy process (AHP) method to weigh the indicators of the comprehensive evaluation of the transfer effect of subway stations and quantified a transfer comfort index in the evaluation system as the per capita transfer space area, which reflected the degree of crowding in a transfer station. Wang et al. [11] used the analytic hierarchy process (AHP)

method to establish five indicators, i.e., transfer time, transfer distance, coordination, comfort, and safety, to comprehensively evaluate the transfer efficiency of subway stations. In this system, the transfer space is only quantitatively described by the transfer distance index value [12]. The analytic hierarchy process (AHP) is a systematic evaluation method combining qualitative and quantitative aspects that has certain advantages in dealing with complex decision-making problems. Therefore, this study adopted the field investigation and analytic hierarchy process (AHP) method to establish the environmental suitability evaluation of transfer spaces in subway stations.

Passengers are easily affected by environmental intervention in subway transfer spaces [13]. Han et al. measured the physical environment of six subway stations in Seoul and conducted a survey of 5282 passengers. The questionnaires considered sound, light, thermal environment, and overall comfort [14]. Zhu et al. analyzed the potential influencing factors of passenger transfer flow based on a nested logit model [15]. Katie proposed that passengers have different requirements for transfer space comfort based on different travel purposes [16]. Jing paid attention to factors such as safety and lighting in their research and conducted a demand analysis [17]. Jiang et al. proposed that transfer efficiency is affected by factors such as air quality and the thermal environment in a station [18]. According to Brighton's hypothesis, Liu et al. divided the transfer distance into five levels: very ideal, ideal, acceptable, tolerable, and intolerable [19]. Wu et al. investigated the particulate matter in Beijing subway transfer stations in China and used the relative warmth index (RWI) to evaluate the thermal comfort of passengers during the transfer process [20]. In addition, Hoeven et al. [21] analyzed and summarized the rules of "satisfactory" stations in nine European cities' subway stations, and they pointed out that factors such as underground morphology, visibility, capacity, and proximity are necessary conditions that affect passengers' transfer efficiency. The suitability of urban subway transfer spaces is affected by many factors such as facilities, environment, and space. A transfer space environment should meet the physiological and psychological needs of passengers by creating a good spatial perception that is appropriate to passengers' behavioral preferences [22,23].

Although domestic and foreign scholars have conducted a lot of research on the environmental quality, space design, and transfer efficiency of urban subway transfer spaces, the improvement of a single influencing factor is not enough to systematically improve the environmental quality of transfer spaces. Different users have different needs for transfer spaces, and a systematic evaluation of current standards is necessary to improve the environmental quality of such spaces.

Therefore, in order to improve the travel experience of passengers in transfer spaces and improve the efficiency of transfer, based on the passengers' needs and industry standards, this research established a system for evaluating the environmental suitability of a transfer space with five primary indicators (safety, convenience, practicability, comfort, and aesthetics) and 17 secondary indicators. The fuzzy comprehensive evaluation method was used to evaluate the transfer space environments of eight sample subway stations in Shanghai with passengers as the main body. The linear relationship between the indicators was analyzed. Finally, the evaluation results were used to propose corresponding optimization strategies to provide scientific references for improving the environmental quality of urban subway transfer spaces.

## 2. Research Samples and Data Collection

### 2.1. Research Samples

The Shanghai subway has a large number of transfer stations, and its transfer lines are complex and diverse. Therefore, the transfer spaces of Shanghai subway stations were selected as the research object of this study, and the evaluation samples were chosen to represent typical transfer spaces of four-line, three-line, and two-line intersection stations in order to make the evaluation results more objective. After screening, Century Avenue Station with four-line cross-transfer; People's Square Station, Shanghai Railway Station,

Hanzhong Road Station, and Xujiahui Station with three-line cross-transfer; and East Nanjing Road Station, Jiangsu Road Station, and Laoximen Station with two-line cross-transfer were selected as the research samples for this review (Figure 1 and Table 1).

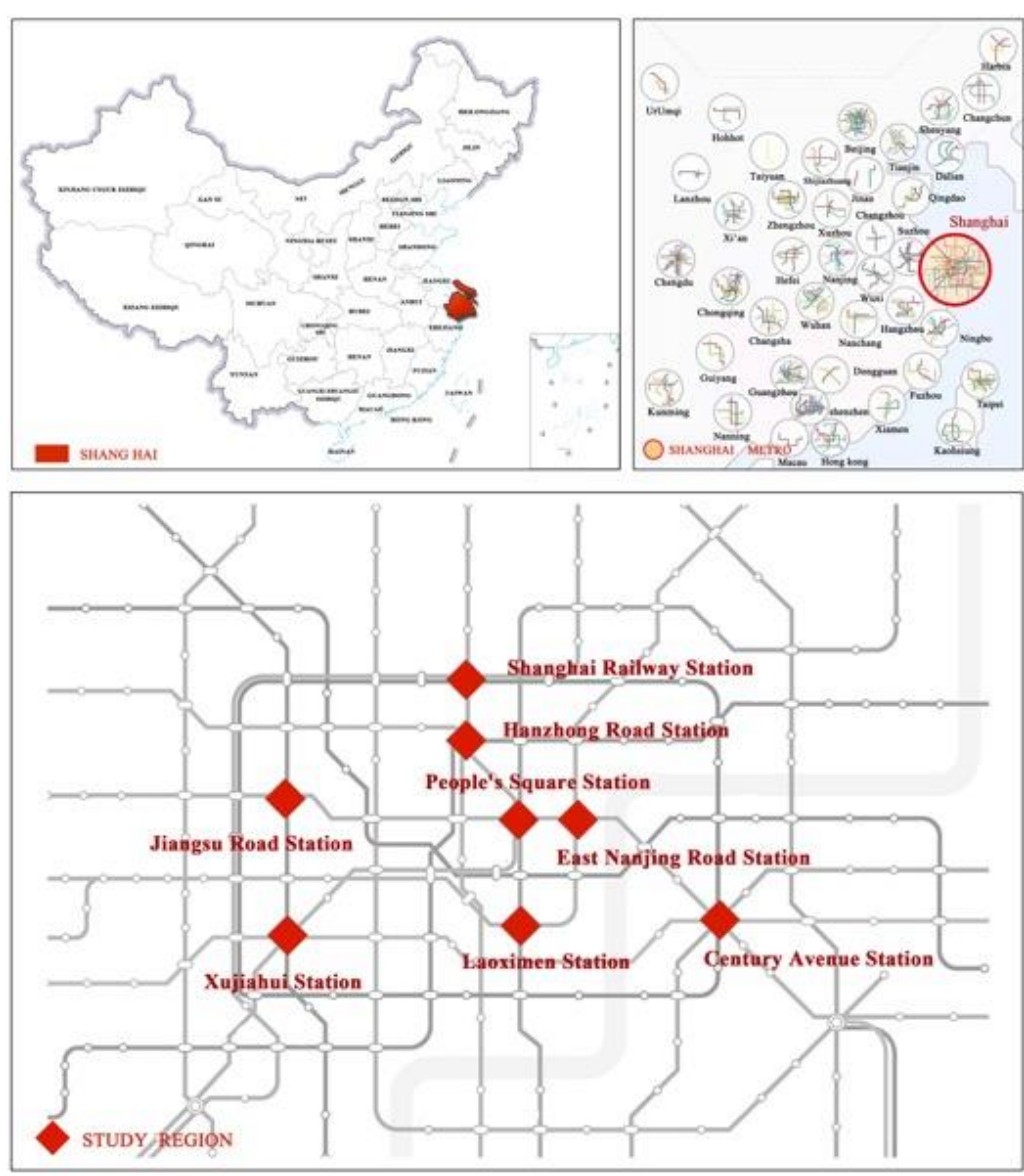

**Figure 1.** Spatial location map of research samples.

**Table 1.** Overview of the research samples.

| Station Name | Century Avenue Station (CAS) | People's Square Station (PSS) | Shanghai Railway Station (SRS) | Hanzhong Road Station (HZRS) |
|---|---|---|---|---|
| Construction time | 2000/2009 | 1995/2000/2007 | 1995/2000/2005 | 1995/2015 |
| Transfer line | 4-line transfer | 3-line transfer | 3-line transfer | 3-line transfer |
| | 2, 4, 6, 9 | 1, 2, 8 | 1, 3, 4 | 1, 12, 13 |
| Station size (Platform width × Station length) | (12 × 269) | (14 × 358) | (24 × 595) | (22 × 214) |

**Table 1.** *Cont.*

| Station Name | Century Avenue Station (CAS) | People's Square Station (PSS) | Shanghai Railway Station (SRS) | Hanzhong Road Station (HZRS) |
|---|---|---|---|---|
| Station form | Underground Island Type | Underground Island Type (1/2) and Island Side Type (8) | Underground Island Type (1) and Above Ground Island Type (3/4) | Underground Island Type |
| |  |  |  |  |
| Transfer method | Station hall transfer | Combined transfer | Passage transfer | Combined transfer |
| |  |  |  |  |
| Transfer time/min | 7 | 3.6 | 9.5 | 5 |
| Transfer distance/m | 490 | 250 | 665 | 350 |
| Number of entrances and exits | 8 | 20 | 6 | 8 |
| Average daily passenger flow | 32.9 Ten thousand visits | 26.1 Ten thousand visits | 20.4 Ten thousand visits | 14.2 Ten thousand visits |
| Land-use function around the station | Commercial business and residential | Commercial business, park green space, and residential | Commercial business, transportation facilities, and residential | Commercial business, park green space, and residential |
| |  |  |  |  |
| Station name | Xujiahui Station (XJHS) | East Nanjing Road Station (ENRS) | Jiangsu Road Station (JRS) | Laoximen Station (LXMS) |
| Construction time | 1993/2009/2013 | 2000/2010 | 2000/2009 | 2007/2010 |
| Transfer line | 3-line transfer | 2-line transfer | 2-line transfer | 2-line transfer |
| | 1, 9, 11 | 1, 14 | 2, 11 | 8, 10 |

**Table 1.** *Cont.*

| Station Name | Century Avenue Station (CAS) | People's Square Station (PSS) | Shanghai Railway Station (SRS) | Hanzhong Road Station (HZRS) |
|---|---|---|---|---|
| Station size (Platform width × Station length) | (21 × 205) | (14 × 279) | (12 × 268) | (12 × 203) |
| Station form | Underground Island Type | Underground Island Type (1) and Underground side type (14) | Underground Island Type | Underground Island Type |
| |  |  |  |  |
| | Combined transfer | Passage transfer | Passage transfer | Platform stair transfer |
| Transfer method |  |  |  |  |
| Transfer time/min | 7.5 | 3.5 | 3 | 1.7 |
| Transfer distance/m | 525 | 245 | 210 | 120 |
| Number of entrances and exits | 19 | 7 | 8 | 7 |
| Average daily passenger flow | 18.3 Ten thousand visits | 20.3 Ten thousand visits | 9.9 Ten thousand visits | 12.2 Ten thousand visits |
| Land-use function around the station | Commercial business, education and research, and residential | Commercial business and residential | Commercial business, residential, and education and research | Commercial business and residential |
| |  |  |  |  |
| Legend | Commercial, business and facility land · Administrative office space · Traffic venues and facilities · Sports venue · Cultural, exhibition and facilities sites · Education and research land · Green space and plaza land · Heritage Site · Construction land · Logistics warehousing land · Industrial land · Park land · Residential land · Medical and health land · Territorial waters · Free space | | | |

## 2.2. Data Collection

In order to ensure the authenticity of the data source, a combination of various research methods such as actual measurements of transfer routes, activity observation, and questionnaire interviews were used to calculate the times of different transfer routes for each sample

based on the average field walking values of various types of people. The accuracy of the research results was verified with data comparison. In the actual measurements of transfer route walking, five experimenters walked and transferred at eight transfer stations during morning and evening peaks and normal time periods, the time spent in walking during these periods was measured, and the questionnaire was then filled in. Activity observation was used to observe the activity phenomena of passengers on the spot at the four types of subway transfer stations before filling in the questionnaire. The activity observation locations were set at the entrance and exit, passage, transfer elevator, transfer hall, and platform. Questionnaire interviews were divided into two types: online and offline. The online questionnaire was filled out by scanning a QR code on the Questionnaire Star APP, and the questionnaire was distributed to WeChat groups (WeChat is the most commonly used communication application in China) of different enterprises, communities, and college students in Shanghai. The offline questionnaire used two methods, i.e., open and semi-open interviews, to obtain data, and interviews were conducted face-to-face while respondents were waiting on a platform for the subway. Eighty valid questionnaires were collected from each station, with a total of 640 questionnaires. The respondents included minors under the age of 18, young people between the ages of 18 and 44, middle-aged people between the ages of 45 and 59, and elderly people over the age of 60 (Figure 2). In this survey, young and middle-aged people accounted for 57% of the total, and more than half of the respondents made transfers at the site at least 4 days a week, indicating that they were familiar with the internal environment of the transfer space and could guarantee information significance.

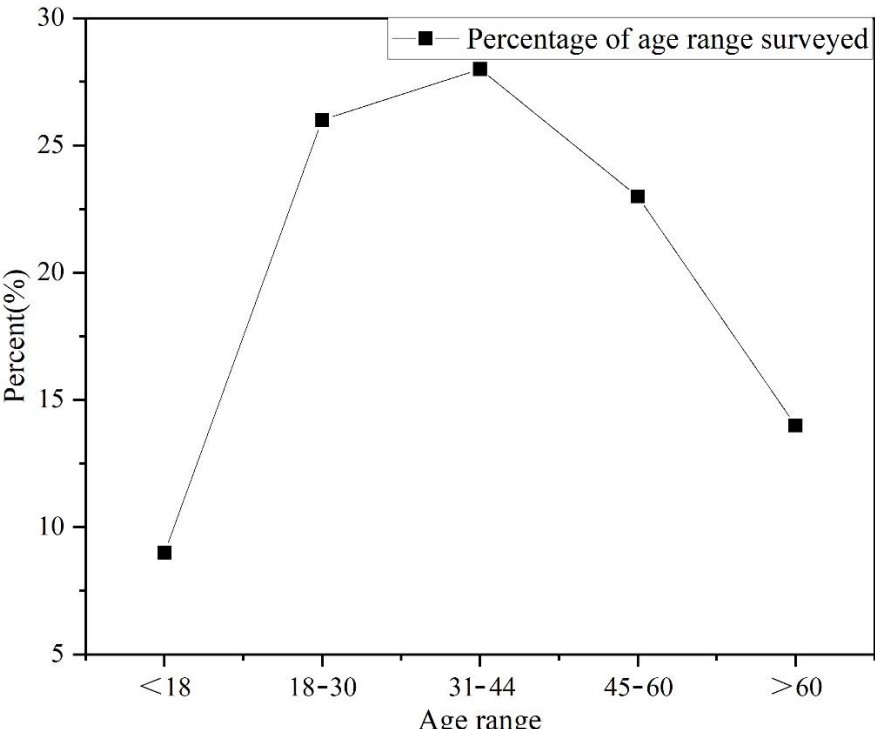

**Figure 2.** Percentage chart of respondent age distribution.

## 3. Construction of Evaluation Index System

### 3.1. Extraction of Evaluation Indicators

Based on the theoretical research and normative standards of subway transfer spaces and the actual research method of unstructured interviews, we gained an understanding of the physical and psychological needs of passengers of different ages, occupations, and transfer frequencies for transfer space environments, obtained information on potential influencing factors of the environmental suitability of transfer spaces, and established

an initial evaluation system. In order to improve the scientificity and credibility of the evaluation system, we consulted relevant scholars and experts of the Metro Design Institute and underground space and environmental design majors for professional opinions, and then we further adjusted and optimized the initially established evaluation index system. When the selection rate of an evaluation factor was less than 60%, the evaluation factor was determined to be invalid. Finally, the evaluation of the environmental suitability of subway transfer spaces was summarized into five evaluation indicators of safety, practicality, convenience, comfort, and aesthetics, with a total of 17 secondary evaluation indicators (Table 2).

**Table 2.** Evaluation indicators and descriptions of the environmental suitability of transfer spaces in urban subway stations.

| Target Layer | Criterion Layer | Evaluation Factor Layer | Indicator Description |
|---|---|---|---|
| Evaluation system for environmental suitability of urban subway transfer space P | Safety $P_1$ | Safety facilities $P_{11}$ [24] | Emergency call facilities, fire hydrant facilities, emergency information lights, etc. |
| | | Daily safety control $P_{12}$ [25] | Update and maintenance of facilities, anti-slip measures on the ground in rainy days, and control of the flow of people by means of closing elevators, setting railings, setting duty officers at important nodes, etc. |
| | | Evacuation emergency safety $P_{13}$ [26] | A relevant escape system used to escape from emergency disasters and ensure the rapid and effective evacuation of people to the above ground space. |
| | Convenience $P_2$ | Guide signs $P_{21}$ [27] | Guiding information is easy to identify and remember. The layout of the guide facilities is convenient for finding transfer information. Multi-form combination of ground guide, wall guide, hanging guide, etc. |
| | | Transfer route $P_{22}$ [28,29] | Whether the transfer route is simple, the degree of winding, and the number of turns at the platform. |
| | | Transfer time $P_{23}$ [30] | Time taken to reach the transfer platform. |
| | Practicality $P_3$ | Leisure activity space $P_{31}$ [31] | Organize subway cultural corridors, art exhibitions, public welfare-themed activities, etc. |
| | | Commercial behavior space $P_{32}$ [32] | Including shops selling snacks, beverages, and souvenirs, as well as providing convenience consumer goods. |
| | | Service facilities $P_{33}$ [33] | Facilities such as toilets, vending machines, maternity rooms, ATMs, lost and found offices, drinking fountains, and suggestion boxes. |
| | | Accessibility facilities $P_{34}$ [34] | Equipment for the visually, hearing, and mobility impaired, as well as the elderly, children, pregnant women and other special passengers. |
| | | Access facilities $P_{35}$ [35] | The location, number and size of passages. The location and number of stairway escalators to meet the needs of use. |

**Table 2.** *Cont.*

| Target Layer | Criterion Layer | Evaluation Factor Layer | Indicator Description |
|---|---|---|---|
| | Comfort $P_4$ | Thermal comfort $P_{41}$ [36,37] | Whether the temperature, humidity, fresh air, etc., are comfortable. |
| | | Light comfort $P_{42}$ [38] | Requirements for contrast, color temperature, and brightness in different time periods (peak and off-peak) and different places (shops, platforms, etc.) in the same place. |
| | | Sound comfort $P_{43}$ [39] | The degree of noise impact of the train entering and leaving the station, the flow of people, and the equipment in the subway station. Whether the broadcast, emergency prompts, telephone ringtones, etc., can be clearly heard. |
| | Aesthetics $P_5$ | Decoration $P_{51}$ [40] | Overall color matching of space, regional cultural characteristics, and artistic design. |
| | | Natural landscape $P_{52}$ [41] | Introduction of green vegetation and water landscape. |
| | | Environmental hygiene $P_{53}$ [42,43] | Cleanliness of environmental sanitation. |

*3.2. Evaluation Method and Process*

3.2.1. Evaluation Method

An environmental suitability evaluation index should consider the relationship between "people, behavior, and environmental needs", and evaluation indexes should be judged based on both the subjective perception and objective environments. Therefore, the Delphi [44] and analytic hierarchy process (AHP) [45] methods were combined to determine the weights of the evaluation indexes in our study, and the fuzzy comprehensive evaluation method was used to evaluate the environmental suitability of subway station transfer spaces.

3.2.2. Construction of Evaluation Process

1.  Construction of the recursive hierarchy model

The selection of the evaluation indexes followed the principles of comprehensiveness, scientific nature, simplicity, and human-centeredness. Accordingly, a three-layer progressive hierarchy consisting of a target layer, criterion layer, and evaluation factor layer was established for the research content (Figure 3).

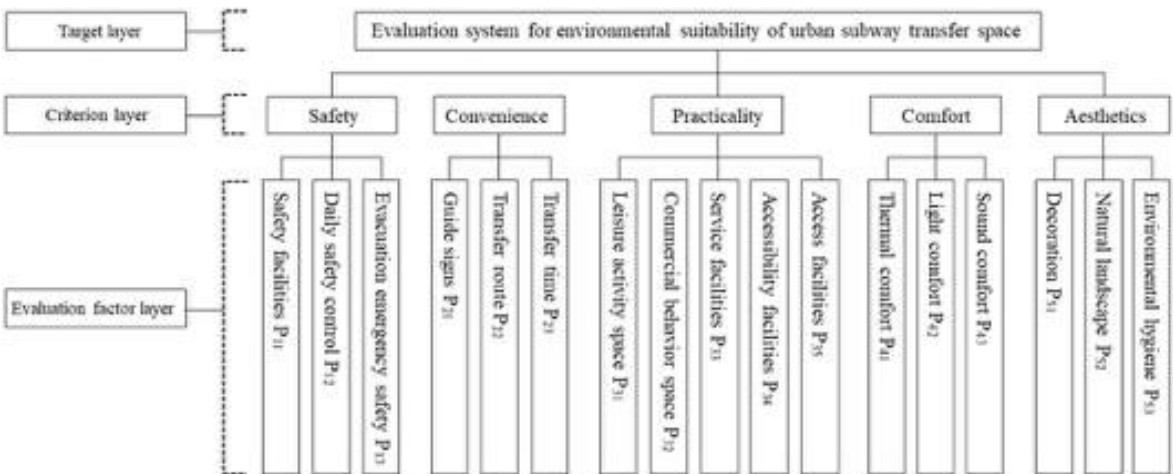

**Figure 3.** Evaluation model of the environmental suitability of transfer spaces in urban subway stations.

Target layer: Evaluation of environmental suitability of urban subway transfer space.
Criterion layer: Five indicators: safety, practicality, convenience, comfort, and aesthetics.
Evaluation factor layer: A total of 17 evaluation indicators corresponding to the criterion level.

2.   Matrix index weight assignment

In this index weight assignment process, T. L. Satty's 1-9 scaling method [46] was used and applied to the investigation process of the expert weight questionnaire to quantify the fuzzy weight judgment of the evaluation index.

3.   Construction of a judgment matrix

The geometric mean was used to determine the mean of the experts' scores, and hierarchical analysis was used to determine the matrix in pairs, which can be expressed as follows:

$$E = |A_1, A_2, \ldots\ldots, A_N| \tag{1}$$

$$w = \begin{array}{c} A_1 \\ A_2 \\ \vdots \\ A_N \end{array} \begin{bmatrix} 1 & A_{12} & \cdots & A_{1N} \\ A_{21} & 1 & \cdots & A_{2N} \\ \vdots & \vdots & \ddots & \vdots \\ A_{N1} & A_{N2} & \cdots & 1 \end{bmatrix} = (A_{ij})_{N \times N}, (A_{ij} = 1/A_{ji}) \tag{2}$$

Equations (1) and (2) were used to determine the importance order of the constituent elements of each index layer, where $E$ is the total evaluation index, $w$ is the element of each index layer, and the importance of the index layer element relative to itself is 1.

With respect to the sub-element set $A_i$, the sub-element set $A_i = |A_{i1}, A_{i2}, \ldots\ldots, A_{ik}|$ in each indicator layer element was compared in pairs to obtain the importance judgment matrix as follows:

$$A_i = |A_{i1}, A_{i2}, \ldots\ldots, A_{ik}| \tag{3}$$

$$w_i = \begin{array}{c} A_{i1} \\ A_{i2} \\ \vdots \\ A_{ik} \end{array} \begin{bmatrix} 1 & a_{12}^i & \cdots & a_{1k}^i \\ a_{21}^i & 1 & \cdots & a_{2k}^i \\ \vdots & \vdots & \ddots & \vdots \\ a_{k1}^i & a_{k2}^i & \cdots & 1 \end{bmatrix} = a_{k \times k}^i, i = 1, 2, \cdots k, \left(a_{ij}^i = 1/a_{ji}^i\right) \tag{4}$$

In Equations (3) and (4), $a_{ij}^i$ represents the discriminant value of the relative importance of element $A_{ii}$ to $A_{ij}$ in each index of the criterion layer $A_i$. The importance order of each index can be determined by calculating the ranking of the judgment value of the elements of the index layer.

4. Determination of the relative weights of each indicator in the criterion layer and the indicator layer

According to the results of the judgment matrix, the importance ranking of each index was obtained. The product square root method was used to solve the judgment matrix, and then the relative weights of the elements under the single criterion $M$ and $A_i$ were obtained.

The geometric mean of each row of the judgment matrix can be calculated by using the product square method $(\overline{W}_i)$:

$$\overline{W}_i = \left( \prod_{j=1}^{n} a_{ij} \right)^{\frac{1}{n}}$$
$$ij = 1, 2, \ldots \ldots, n \tag{5}$$

where $a_{ij}$ is the *j-th* element in the *i-th* layer of the original index layer importance judgment matrix, n is the number of indicators of each index layer under the criterion layer, and $\overline{W}_i$ is the geometric mean of the *i-th* layer in the original index layer importance judgment matrix.

The geometric mean of each row can be normalized to obtain the feature vector:

$$W_i = \frac{\overline{W}_i}{\sum_{j=1}^{n} \overline{W}_j}$$
$$ij = 1, 2, \ldots \ldots, n \tag{6}$$

where $W_i$ represents the weight of the *i-th* index layer index of the original index layer importance judgment matrix, n represents the number of indicators in the original indicator layer, and $\overline{W}_i$ represents the geometric mean of the *i-th* layer of the importance judgment matrix in the original indicator layer.

The largest eigenvalue of the judgment matrix $\lambda_{\max}$ can be calculated as follows:

$$\lambda_{\max} = \frac{1}{n} \sum_{i=1}^{n} \frac{(\sum_{j=1}^{n} a_{ij} W_j)}{W_i}$$
$$ij = 1, 2, \ldots \ldots, n \tag{7}$$

The consistency index $CI$ and the consistency ratio $CR$ can be calculated as follows:

$$CI = \frac{\lambda_{\max} - n}{n - 1}$$
$$CR = \frac{CI}{RI} \tag{8}$$

In Equation (8), when $n$ = 2, the positive and negative results of the matrix are consistent, so there is no need to verify the consistency of the judgment matrix. When $n > 2$, the function of the $CR$ calculation matrix is to judge whether the results are consistent. $CR$ (Consistency Ratio) = $CI$ (Consistency Index)/$RI$ (Random Consistency). The values of $RI$ (random consistency) are shown in Table 3.

**Table 3.** Average random consistency index.

| Order | 1 | 2 | 3 | 4 | 5 | 6 | 7 | 8 | 9 |
|-------|---|---|------|------|------|------|------|------|------|
| *RI* | 0 | 0 | 0.52 | 0.89 | 1.12 | 1.24 | 1.32 | 1.41 | 1.45 |

$CI$ is an indicator for judging the consistency of the matrix when the following equation is satisfied:

$$CR = \frac{CI}{RI} < 0.10 \tag{9}$$

It was considered that the judgment matrix met the consistency condition. On the contrary, the judgment matrix model needed to be appropriately adjusted. According to the above method, the matrix of each level and the consistency of weights were calculated.

Using the above method, the consistency test, combination, and calculation of the weights of indicators at each level were carried out, and the final weight coefficients were obtained, as shown in Table 4.

**Table 4.** Summary of weights of indicators at all levels.

| Target Layer | Criterion Layer | Combined Weights | Evaluation Factor Layer | In-Group Weight | Final Combined Weight |
|---|---|---|---|---|---|
| Evaluation system for environmental suitability of urban subway transfer space P | Safety P$_1$ | 0.3968 | Safety facilities P$_{11}$ | 0.2000 | 0.0794 |
| | | | Daily safety control P$_{12}$ | 0.2000 | 0.0794 |
| | | | Evacuation emergency safety P$_{13}$ | 0.6000 | 0.2381 |
| | Convenience P$_2$ | 0.2908 | Guide signs P$_{21}$ | 0.2000 | 0.0582 |
| | | | Transfer route P$_{22}$ | 0.2000 | 0.0582 |
| | | | Transfer time P$_{23}$ | 0.6000 | 0.1745 |
| | Practicality P$_3$ | 0.1705 | Leisure activity space P$_{31}$ | 0.0433 | 0.0074 |
| | | | Commercial behavior space P$_{32}$ | 0.0795 | 0.0136 |
| | | | Service facilities P$_{33}$ | 0.1534 | 0.0261 |
| | | | Accessibility facilities P$_{34}$ | 0.2911 | 0.0496 |
| | | | Access facilities P$_{35}$ | 0.4327 | 0.0738 |
| | Comfort P$_4$ | 0.1011 | Thermal comfort P$_{41}$ | 0.5247 | 0.0531 |
| | | | Light comfort P$_{42}$ | 0.3338 | 0.0338 |
| | | | Sound comfort P$_{43}$ | 0.1416 | 0.0143 |
| | Aesthetics P$_5$ | 0.0408 | Decoration P$_{51}$ | 0.1062 | 0.0043 |
| | | | Natural landscape P$_{52}$ | 0.2605 | 0.0106 |
| | | | Environmental hygiene P$_{53}$ | 0.6333 | 0.0258 |

### 3.2.3. Analysis of Evaluation Weights

According to the results of experts' scoring on the weights of indicators at all levels, P1 was the most important factor affecting the environment suitability of a transfer space, followed by P2, P3, and P4, and P5 with lower degrees of influence.

The final combined weight results (Figure 4) showed that the importance of P13 and P23 in the secondary influencing factors was much greater than that of other index factors, and they were found to have the greatest influence on the score of passengers' satisfaction with the environmental suitability of the transfer space. The weight values of P11, P12, and P35 were found to be relatively balanced, the weight values of P22 and P21 were the same, and the weight values of P41, P34, P42, P33, and P53 were also higher. Therefore, it was necessary to pay more attention to these influencing factors when carrying out the single-item design.

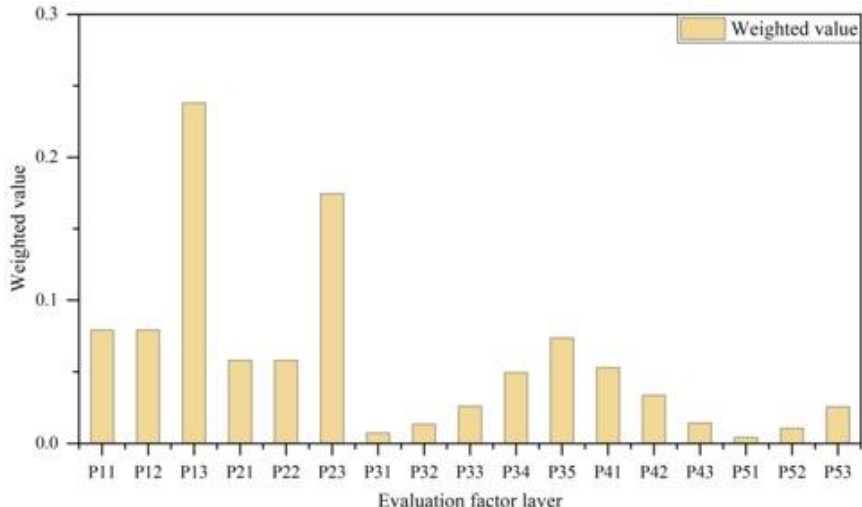

**Figure 4.** Final combined weights.

### 3.3. Current Suitability Evaluation

According to the evaluation indicators and methods, the environmental suitability of the transfer spaces of Shanghai subway stations was evaluated.

First of all, Construction of an Evaluation Index Set M (Equation (10)).

$$
\begin{aligned}
P &= \{P_1, P_2, P_3, P_4, P_5\} \\
P_1 &= \{P_{11}, P_{12}, P_{13}\} \\
P_2 &= \{P_{21}, P_{22}, P_{23}\} \\
P_3 &= \{P_{31}, P_{32}, P_{33}, P_{34}, P_{35}\} \\
P_4 &= \{P_{41}, P_{42}, P_{43}\} \\
P_5 &= \{P_{51}, P_{52}, P_{53}\}
\end{aligned}
\tag{10}
$$

In the next place, Construction of an Evaluation Set P. The Likert scale method was used to measure the evaluation statistics of the transfer space environment questionnaire, and the satisfaction level of the space environment was divided into 5 criteria. The evaluation set was: V = {V1, V2, V3, V4, V5} = {good, fairly good, fair, fairly poor, poor} = {5, 4, 3, 2, 1}, and each index element was scored. The evaluation semantic scale is shown in Table 5.

**Table 5.** Average random consistency index.

| Semantics | Good | Fairly Good | Fair | Fairly Poor | Poor |
|---|---|---|---|---|---|
| Semantic score | 5 | 4 | 3 | 2 | 1 |
| Evaluation scope | $F \geq 4.5$ | $4.5 > F \geq 3.5$ | $3.5 > F \geq 2.5$ | $2.5 > F \geq 1.5$ | $1.5 > F$ |

In the next place, Construction of a Weight Vector Set (Equation (11)).

$$
W = (w_1, w_2, \cdots\cdots, w_n)
\tag{11}
$$

In the next place, Establishment of a Fuzzy Evaluation Matrix (Equation (12)).

$$
R = (R_1, R_2, \cdots\cdots, R_n)
\tag{12}
$$

In the next place, Comprehensive Multi-Factor Evaluation Combining Weight Sets (Equation (13)).

$$
E = W \times R = (w_1, w_2, \cdots, w_k) \times
\begin{bmatrix}
r_{11} & r_{12} & \cdots & r_{1n} \\
r_{21} & r_{22} & \cdots & r_{2n} \\
\cdots & \cdots & \cdots & \cdots \\
r_{k1} & r_{k2} & \cdots & r_{kn}
\end{bmatrix}
\tag{13}
$$

The transfer space environments of 8 subway stations in Shanghai were quantified, and the final comprehensive scoring results are shown in Figure 5.

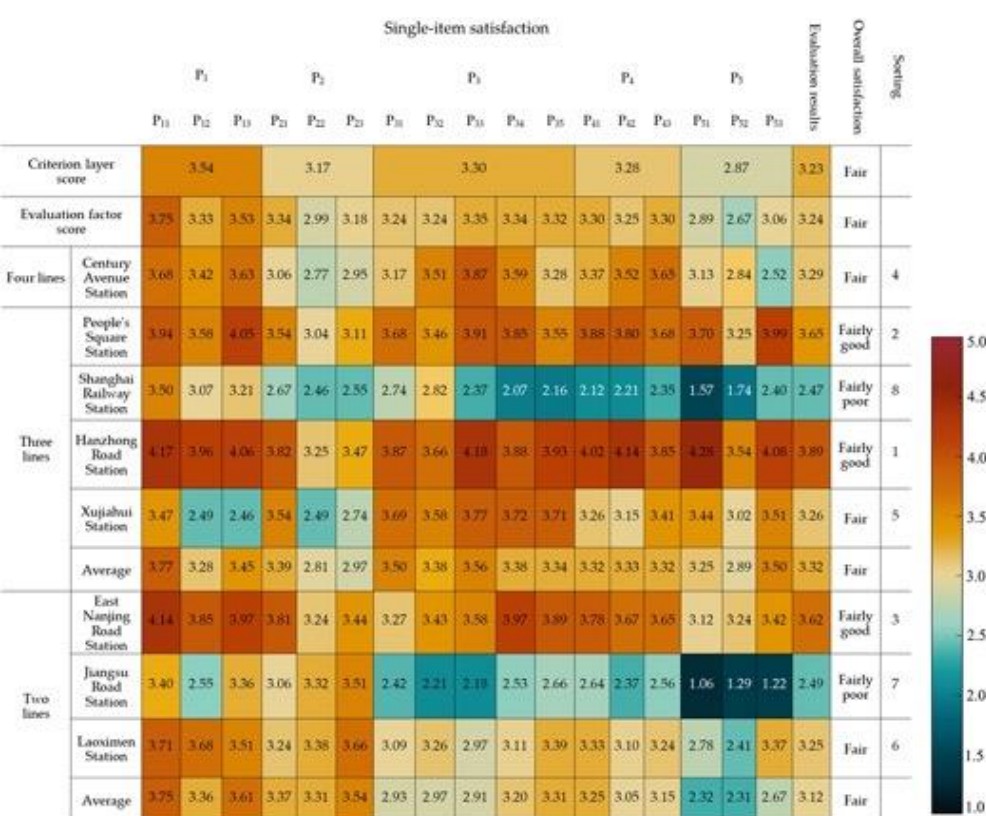

**Figure 5.** Overall and single-item evaluation scores on the factor satisfaction of transfer space environments.

## 4. Result Analysis

### 4.1. Results of the Overall Suitability Evaluation of the Sample Stations

The evaluation results of the transfer space of urban subway stations showed that the overall satisfaction of the respondents with transfer space environments was not satisfactory, and none of the sample stations had a "good" evaluation (Table 5). In the evaluation of different transfer lines, the overall satisfaction of the three-line transfer station (3.32) was slightly higher than that of the two-line transfer station (3.12). In the single-item factor evaluation, the safety satisfaction (3.54) degree of the transfer space environments was the highest, the satisfaction degrees of practicality (3.3) and comfort (3.28) were second, and the satisfaction degrees of convenience (3.17) and aesthetics (2.87) were the lowest (Figure 6). Among the three-level evaluation indicators, passengers were most dissatisfied with transfer time, transfer route, decoration, and natural landscape. At the same time, there were certain differences in the satisfaction of single-item factors among different transfer lines (Figure 7).

### 4.2. Single-Item Factor Suitability Evaluation

#### 4.2.1. Safety

Passengers' safety evaluation value of the sample stations was 3.54, and the result was "fairly good". The safety satisfaction (4.17) of Hanzhong Road Station was the best. The three-level index evaluation result of safety was safety facilities (3.75) > evacuation emergency safety (3.53) > daily safety control (3.33) (Figure 8). The main safety problems were reported to be aging equipment, untimely maintenance of lighting facilities, a lack of monitoring facilities, and stair passages with a high breakage rate, all of which increase the amount of safety hazards in the transfer space. The lack of an estimation of the future flow of people at the beginning of the design, the unreasonable design of the space flow line, and exposed pipelines, among other factors, made passengers feel uneasy (Figure 9).

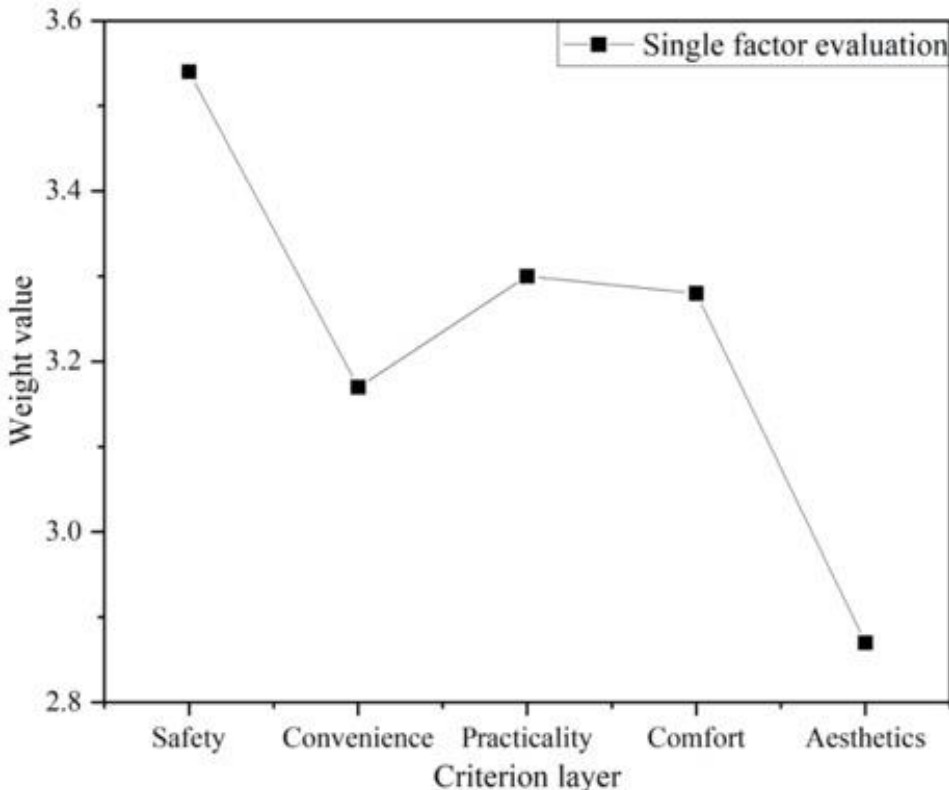

**Figure 6.** Evaluation results of the environmental criterion level of transfer spaces.

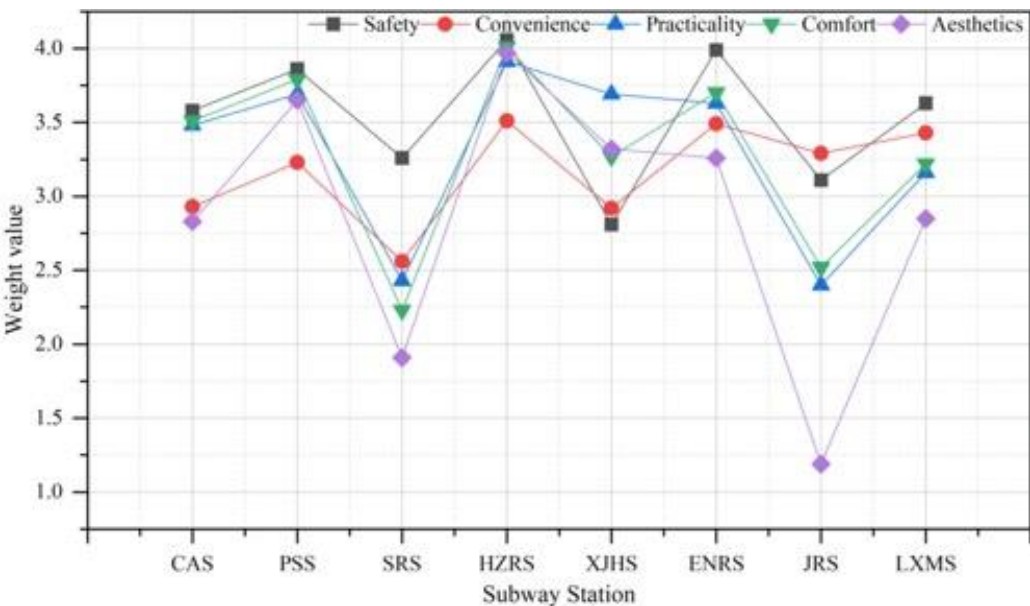

**Figure 7.** Evaluation index results of the transfer space environment criterion level at sample stations.

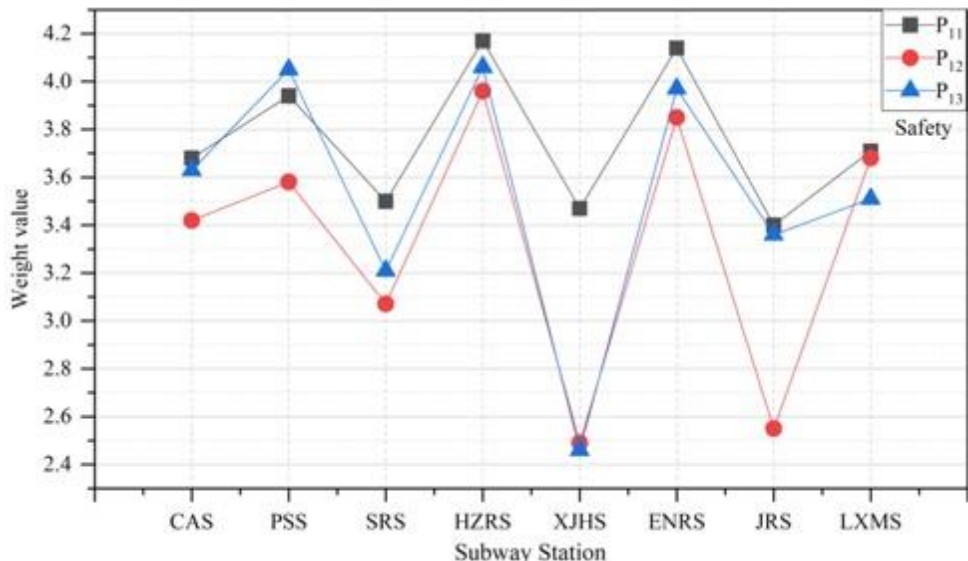

**Figure 8.** The evaluation results of the environmental safety evaluation index of the transfer spaces of sample stations.

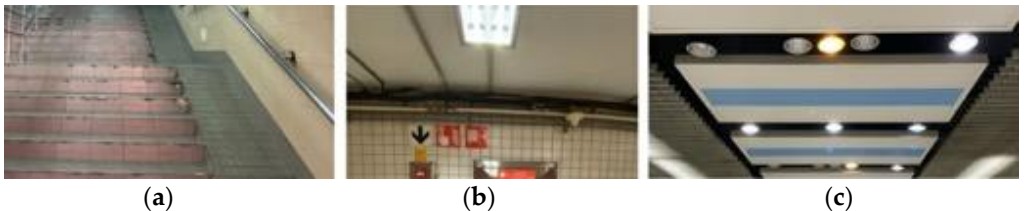

**Figure 9.** Safety status of transfer space environments at sample stations: (**a**) steps of wear; (**b**) pipeline exposure; (**c**) damaged lamps. Images courtesy of the author.

4.2.2. Convenience

Passengers rated convenience at 3.17, with a "fair" result. The evaluation showed that the convenience evaluation factor of the three-line and four-line transfer spaces (3.03) was lower than that of the two-line transfer space (3.4). We found that 65.9% of passengers believed that a simple transfer route, short transfer time, and clear guidance signs were most important (Figure 10). Passengers with clear travel purposes put forward higher requirements for the spatial accessibility and convenience of the transfer spaces. Traveling, shopping, and partying passengers had higher requirements for the convenience of the transfer space environments than those with fixed transfer routes and strong travel purposes.

The main problems at present were reported as follows: ① The information of guide signs is obscured with poor recognition; ② the scale of transfer spaces is not reasonable, the sizes of passageways built more than 15 years ago are too narrow and too low, and passageways of less than 15 years old are too wide and too high, reducing the space for passengers; and ③ two-way mixed traffic in transfer passageways is significant, which makes originally narrow passageways more cramped. As a result, transfer times are prolonged, especially during peak hours (Figure 11).

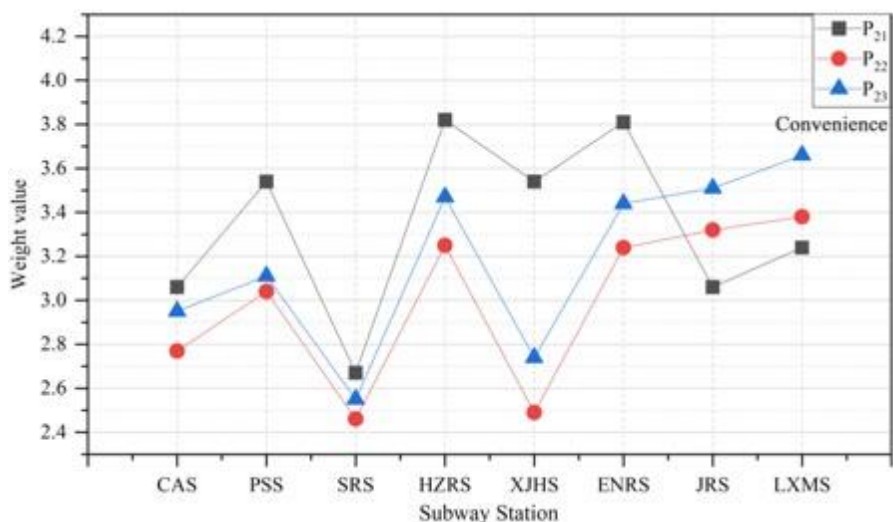

**Figure 10.** The evaluation results of the evaluation index of the convenience of the transfer space environments of sample stations.

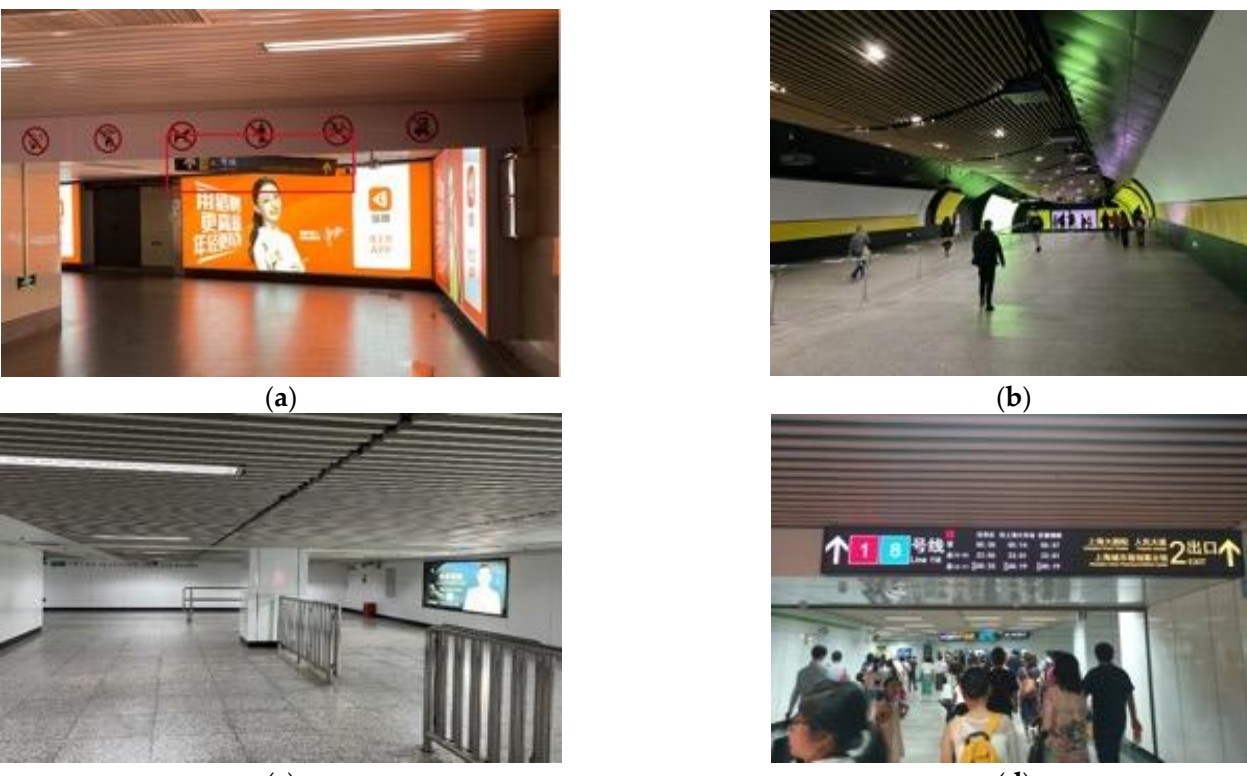

**Figure 11.** Sample station transfer space environment convenience status problem: (**a**) Identification information severely obscured; (**b**) passageway too wide; (**c**) passageway too low; (**d**) two-way mixed traffic. Images courtesy of the author.

### 4.2.3. Practicality

In terms of practicality, the passenger's evaluation value was 3.3, and the "fairly good" stations were Hanzhong Road Station (3.9), Xujiahui Station (3.69), People's Square Station (3.69), and East Nanjing Road Station (3.63). Additionally, the practicability of the transfer space environments of the three-line and four-line transfer stations (3.44) was better than that of the two-line transfer station (3.06) (Figure 12). These results were mainly reflected in the complete range of service facilities, not only common convenience facilities but also

self-service medicine vending machines, self-service umbrella-borrowing machines, and other service facilities (Figure 13). In addition, the practicability of the People's Square Station was found to be more diversified, with a complete range of commercial spaces, as well as leisure and entertainment spaces such as subway music corners and temporary art exhibitions, to meet the needs of passengers for shopping and leisure activities along the way. At the same time, a station transfer space environment with a strong surrounding development intensity and a high rate of land-use mixing was reported to be more practical.

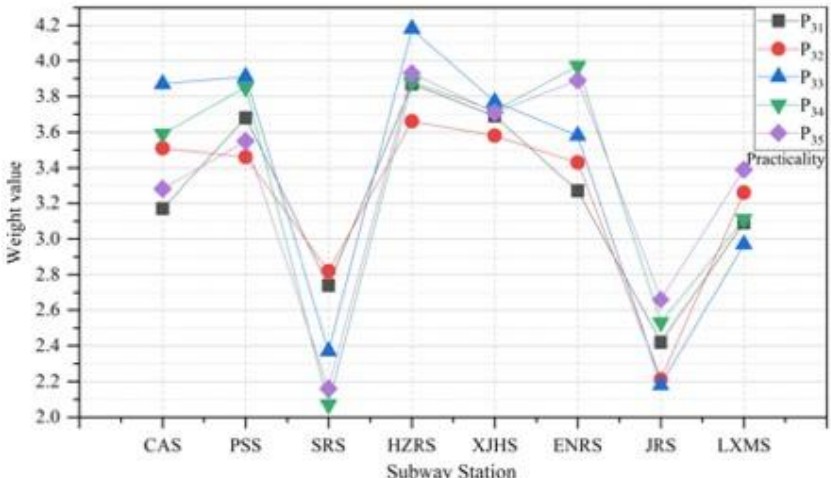

**Figure 12.** Environment practicality evaluation index results of sample station transfer spaces.

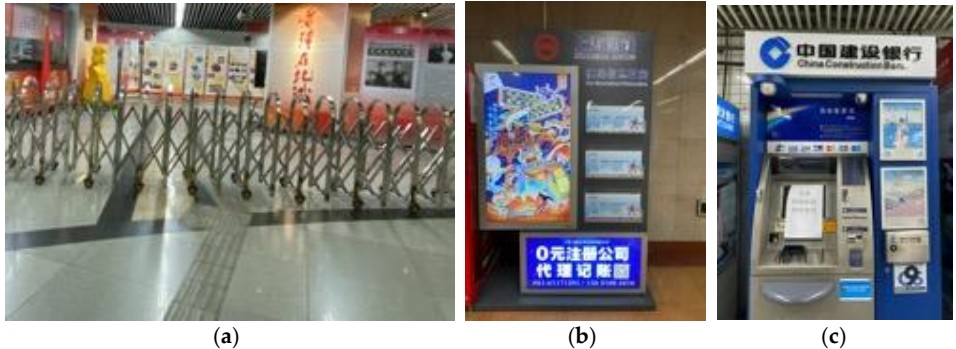

**Figure 13.** The practical situation of transfer space environments at sample stations: (**a**) Blind lane occupation; (**b**) map information without supplemental delivery; (**c**) faulty equipment. Images courtesy of the author.

The current problems of practicability were reported as follows: ① The accessibility facilities in the transfer spaces are not comprehensive enough, the continuity of blind lanes is insufficient, and the blind lanes are seriously occupied; ② it is difficult to find restrooms on station hall floors, and there are no accessible toilets; ③ the maintenance of convenience facilities is not conducted in a timely manner, and there is no supplementary information map of the entire Shanghai rail transit line in the convenience information access column; and ④ the practicability of the two-line transfer space is low, and there are no commercial and cultural entertainment spaces.

### 4.2.4. Comfort

In terms of comfort, the passenger satisfaction value was 3.28, and the light comfort (3.25) score was the lowest among the three-level evaluation factors (Figure 14). Among the evaluation samples, Jiangsu Road Station (2.52) had the lowest score for comfort, which was reflected in the uneven distribution of light environment in the transfer passage. In

addition, there were areas with severe light pollution and dimly lit areas in the passages. The physical properties of the materials of the stations that were constructed earlier were seriously degraded, the equipment was not perfect, and the comfort of older stations was generally lower than that of the newly built stations. The lack of screen doors on waiting platforms was reported to lead to a poor wind environments in station spaces, and the background noise pollution when trains enters stations was reported to be serious, with potential safety hazards (Figure 15).

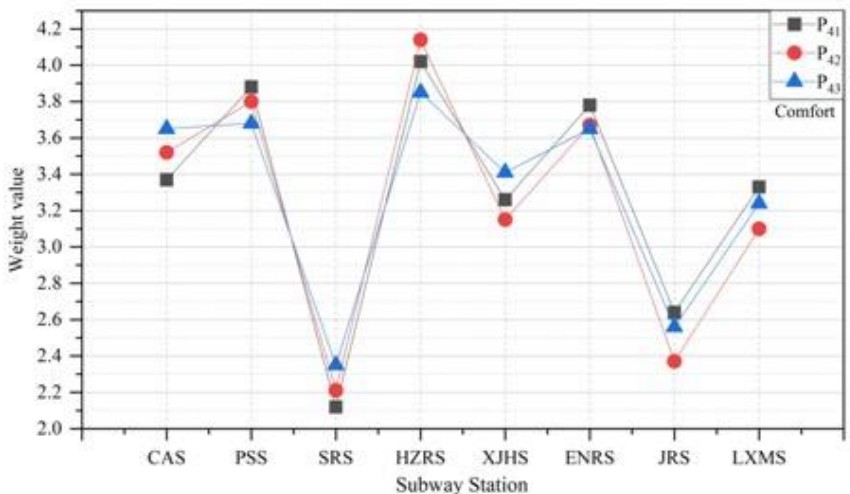

**Figure 14.** The evaluation results of the evaluation index of the environmental comfort of the transfer spaces of sample stations.

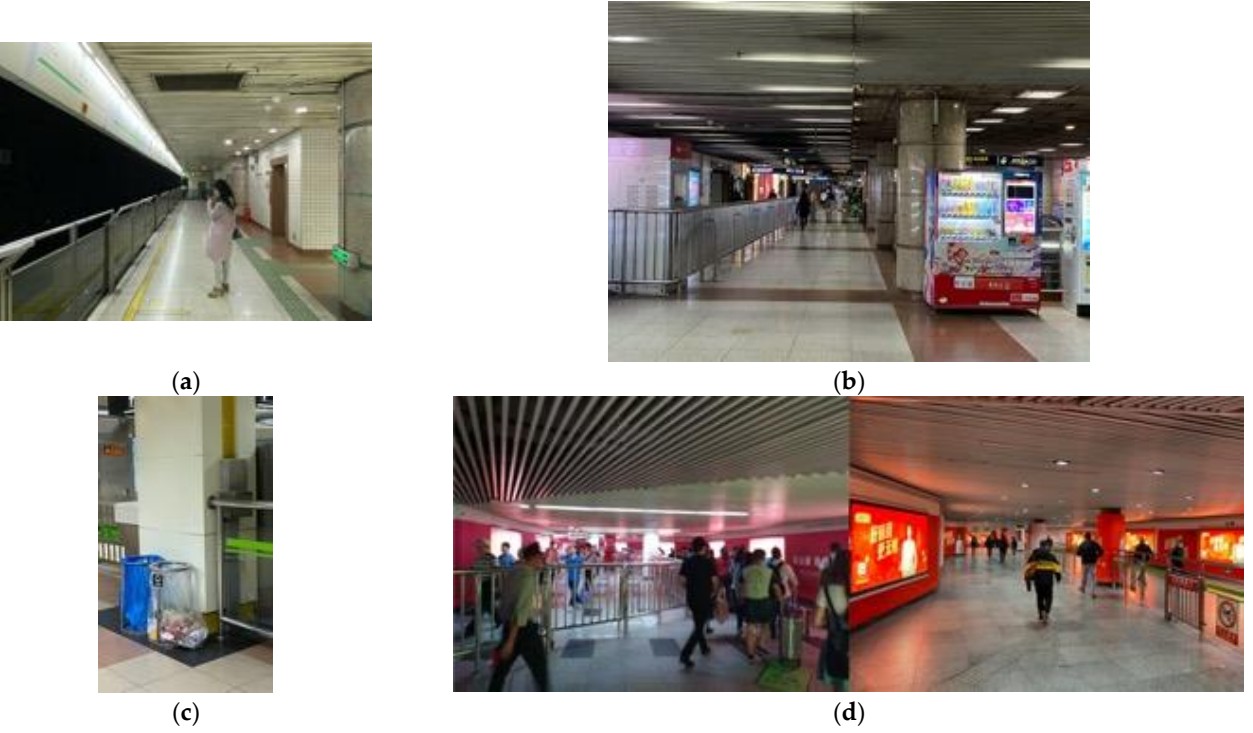

**Figure 15.** Current situation of the transfer space environment comfort of sample stations: (**a**) Poor wind and sound environment; (**b**) aged equipment; (**c**) low sanitation and cleanliness; (**d**) severe light pollution. Images courtesy of the author.

### 4.2.5. Aesthetics

In terms of aesthetics, the passenger satisfaction value was 2.87. The aesthetics had the lowest score of the five first-level indicators. Hanzhong Road Station (3.97) had the highest aesthetics score. The space in the station had a clear theme, the interface was reported to be relevant, and the sanitation environment was good (Figure 16). We found that 75% of the sample stations had the following problems in the aesthetics of the transfer space environment: ① The transfer space environment did not reflect the urban cultural characteristics of Shanghai, the use of cultural elements was not reasonable, and there was little connection with the surrounding environment on the ground; ② the transfer space did not have a uniform color tone, and the interface material and shape were singular and tedious; ③ the indoor sketches were monotonous, and the public facilities had no features; ④ the layout of the lightbox advertisements in the transfer passage was not reasonable enough, and the visual aesthetics was poor; and ⑤ the single-item score of the natural landscape of the eight stations was 2.95, which was the lowest score among the single-item evaluation factors, and there were few green plants in the station spaces (Figure 17).

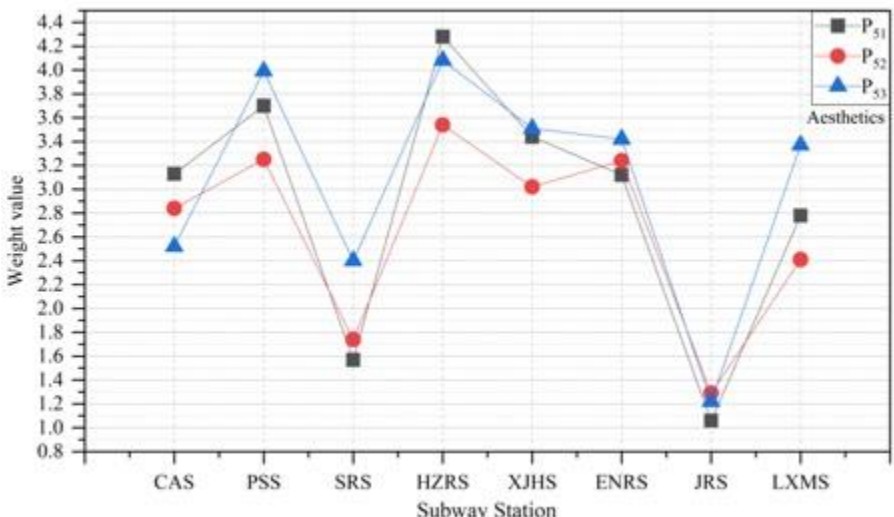

**Figure 16.** The evaluation results of the aesthetic evaluation index of transfer space environments at sample stations.

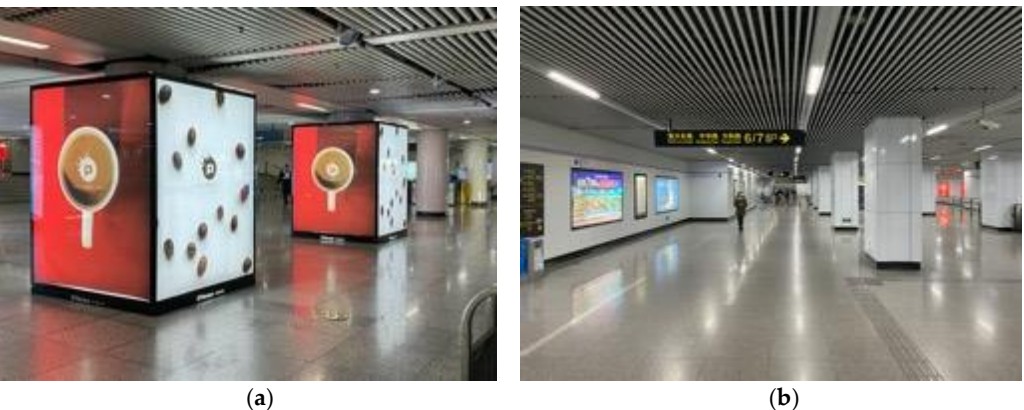

**Figure 17.** Sample station transfer space environment aesthetic status problems: (**a**) Unreasonable advertising layout; (**b**) single and tedious space. Images courtesy of the author.

## 5. Discussions

### 5.1. Comparison of Research on the Evaluation of Transfer Space in Subway Stations

Previous research has mainly focused on underground spaces, and there have been very few comprehensive evaluations of the effects of multiple factors on transfer spaces. Mehta [47] evaluated the vitality and publicity of urban underground spaces from five dimensions, i.e., inclusivity, pleasure, safety, comfort, and interest. Swamidurai [48] established a questionnaire-based structural equation model to evaluate the experience quality of underground spaces regarding the aspects of comfort, functionality, safety, and spatial form. Durmisevic [49] pointed out that functional, psychological, and structural aspects directly affect the quality of entire transfer spaces. In this study, a questionnaire survey combined with a neural network method of space syntax application was proposed to evaluate the safety and comfort of underground transit station spaces. The previous evaluations mainly focused on some index factors and paid more attention to the relationship between a single index and the whole, ignoring the influencing factors between indicators. Previous research has concluded that the main functions of the transfer spaces of subway stations are convenience and practicability. In this researchers, passengers were found to be most concerned with safety in transfer spaces, followed by practicality, comfort, and convenience and aesthetics (Figure 18).

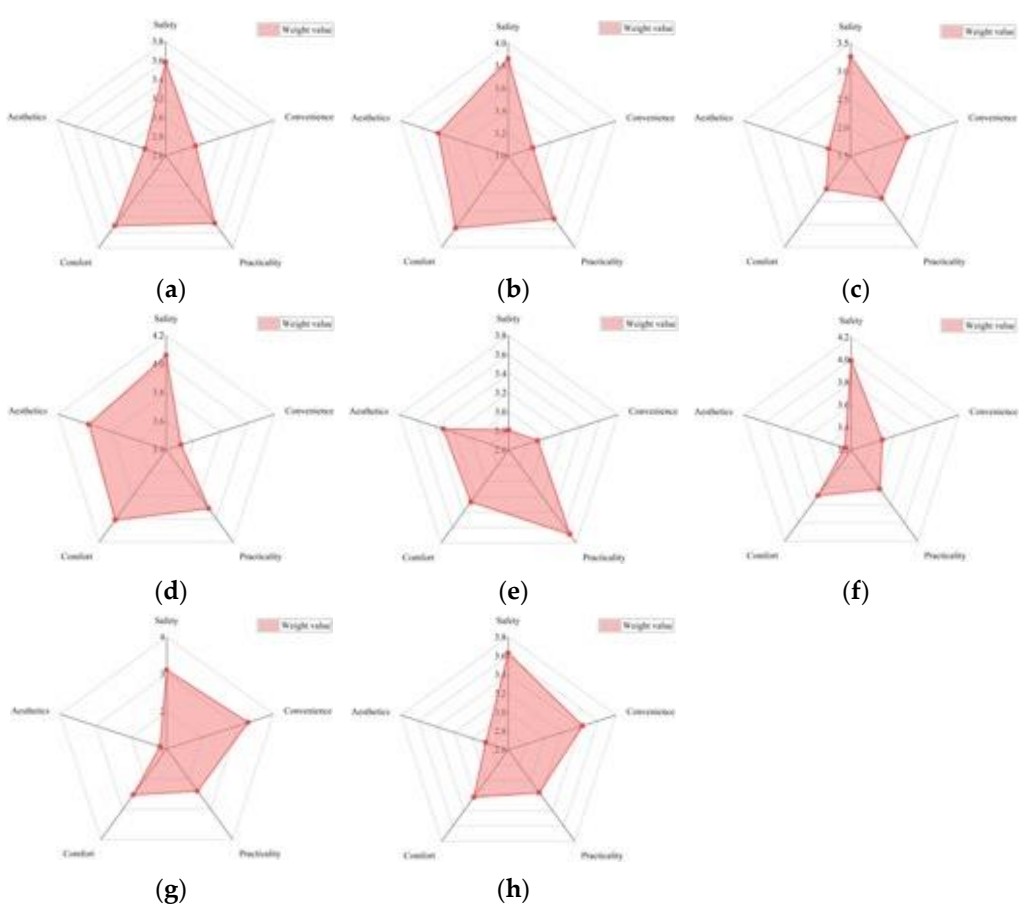

**Figure 18.** Distribution of evaluation results of the transfer environment criterion layer at sample stations: (**a**) Century Avenue Station; (**b**) People's Square Station; (**c**) Shanghai Railway Station; (**d**) Hanzhong Road Station; (**e**) Xujiahui Station; (**f**) East Nanjing Road Station; (**g**) Jiangsu Road Station; (**h**) Laoximen Station.

### 5.2. Correlation of Overall Suitability Evaluation Factors

The closedness of urban subway transfer spaces affects passengers' sense of direction, control over the space, and psychological state, which in turn affect their physical health,

way-finding ability, and behavior in emergencies [5,6,50]. Based on sample transfer spaces of the Shanghai subway, evaluation indicators of the suitability of transfer spaces (safety, convenience, practicability, comfort [51], and aesthetics) were determined in this study. A calculation function was provided to evaluate the suitability of each transfer space and its impact on passengers. Research on the weighting scheme of the suitability factors of the transfer spaces of subway stations showed that passengers were most satisfied with the safety, with a weight of 3.54 for the transfer space environments of subway stations, and they were second-most satisfied with practicability and comfort, with weights of 3.3 and 3.28, respectively. Therefore, these three factors are the key indicators, followed by convenience with a weight of 3.17, and aesthetics with the lowest weight of 2.87.

There were different correlations between various index factors that were found to affect the suitability of transfer spaces by superimposing on each other. Therefore, it is necessary to carry out relevant research on every single factor to improve the environmental quality of transfer spaces. Assessment frameworks for other types of subterranean spaces can also be established using the methods of this study based on existing data. This evaluation method for the suitability of transfer spaces in subway stations can be used as an evaluation tool for this type of space so that designers and passengers can better balance the relationship between suitability indicators [52,53] and formulate a reasonable design or use plan.

We found a positive correlation between the evaluation factor layers in the criterion layer as a whole, that is, between $0.65 < R^2 < 0.95$. Passengers were most concerned with safety, which had positive correlations with various indicators, while the transfer path, transfer time, leisure activity space, commercial behavior space service facilities, and natural landscape indicators were shown to have obvious negative correlations (Figures 19 and 20).

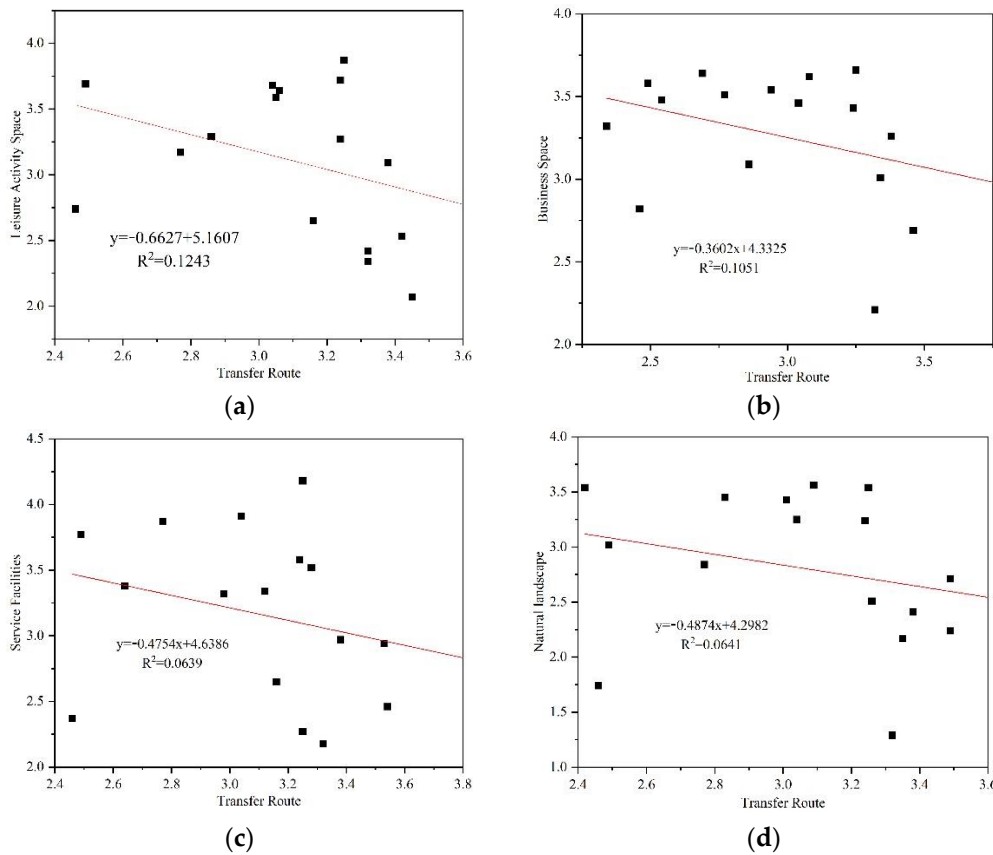

**Figure 19.** (**a**) Relevance of transfer route and leisure activity space; (**b**) relevance of transfer route and business space; (**c**) relevance of transfer route and service facilities; (**d**) relevance of transfer route and natural landscape.

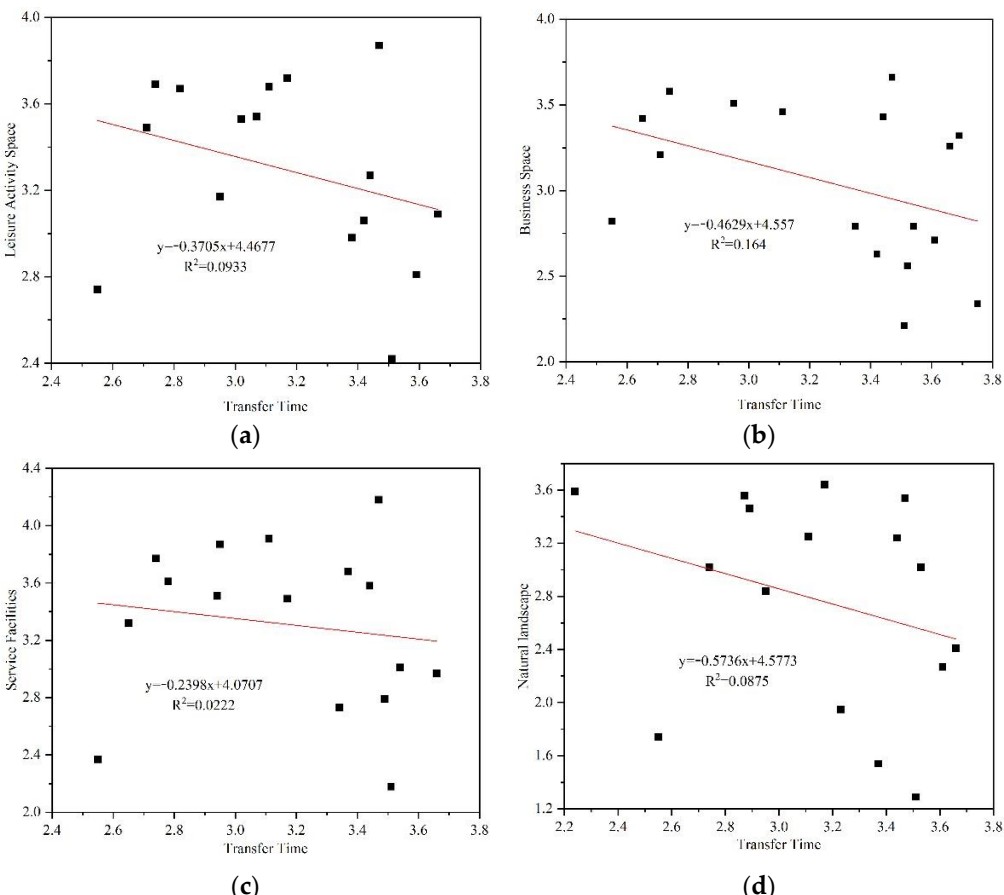

**Figure 20.** (**a**) Relevance of transfer time and leisure activity space; (**b**) relevance of transfer time and business space; (**c**) relevance of transfer time and service facilities; (**d**) relevance of transfer time and natural landscape.

## 6. Conclusions and Future Work

This paper selected the transfer spaces of eight subways in the central urban area of Shanghai as research samples, established an evaluation system for the suitability of urban subway transfer spaces comprising five first-level indicators and 17 second-level indicators, and developed a research method for environmental suitability evaluation. On that basis, the environmental status of the transfer spaces of subway stations was evaluated and analyzed, and the results of this study can provide guidance for future updates of the transfer spaces of subway stations. The main conclusions are as follows.

Among the eight research samples, the first-level index weights were ordered as follows: safety, convenience, practicality, comfort, and aesthetics. Safety had the largest weight, reflecting the fact that safety plays a key role in the experience of transfer behavior. The practicality of the space environment in transfer stations with multi-line intersections and surrounding development functions was stronger than that of two-line cross-transfer stations. The two-line cross-transfer stations had the highest convenience and allowed for better way-finding behavior. The safety and comfort of the transfer spaces at the older stations were poor, and the overall scores of the eight stations were lowest in aesthetics. In addition, there was a strong correlation between indicators, which were found to interact with and influence each other. For example, optimizing the practicability and comfort of a station could improve the safety and convenience of passengers in terms of transfer behavior.

From the perspective of passengers, exploring the relationship between behavior patterns and subway transfer space environments required the establishment of an environmental suitability evaluation method that can be used to quantitatively analyze and

objectively evaluate transfer space environments in different cities based on a single factor or overall systematic evaluation, as well as to discover the problems and potentials of the transfer spaces of different stations, in order to improve the environmental quality of transfer spaces. The evaluation method developed in this study is also applicable to the evaluation of the space suitability of ordinary subway stations, thereby bringing better experience for passengers.

There were still some limitations in the data acquisition process of this study. Due to the random distribution of questionnaires, the participants in this evaluation were mainly young and middle-aged people, and the representativeness of the sampling was insufficient. People with different social attributes have different subjective feelings and needs for transfer space environments due to different travel purposes and transfer routes. The transfer behavior needs of different passengers need to be further explored.

In follow-up research, more accurate and convincing data can be obtained by improving the coverage of evaluators' age composition, education level, etc., and by further conducting investigation and evaluation analyses during holidays, working days, and various periods throughout the day in order to more comprehensively reflect the current situation of transfer space environments. In addition, deep learning technology in the field of artificial intelligence can be applied to study the recognition degree of underground space design features, thereby better providing a scientific basis for the optimal design of subway transfer stations.

**Author Contributions:** Conceptualization, Z.W., X.J., X.Z. and S.T.; data curation, Z.W., X.Z. and S.T.; formal analysis, X.J. and S.T.; funding acquisition, Z.W.; methodology, X.Z., X.J. and S.T.; resources, Z.W., X.J., X.Z. and S.T.; software, Z.W. and S.T.; supervision, X.J.; validation, Z.W.; visualization, Z.W. and S.T.; writing—original draft, Z.W., X.Z. and S.T.; writing—review and editing, Z.W. and S.T. All authors have read and agreed to the published version of the manuscript.

**Funding:** Supported by the National Key R & D Projects in the "13th Five-year Plan" Period: "Research on Evaluation System and Key Technology of Green and Livable Villages and Towns Construction" (2018YFD1100203); Major Research Fund Project of Jiangsu Building Energy Conservation and Construction Technology Collaborative Innovation Center: "Research on Evaluation System of Planning and Construction of Green and Livable Villages and Towns" (SJXTZD2105).

**Institutional Review Board Statement:** Not applicable.

**Informed Consent Statement:** Not applicable.

**Data Availability Statement:** Not applicable.

**Conflicts of Interest:** The authors declare no conflict of interest.

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
