# Peer review of "Research on Environmental Suitability Evaluation of the Transfer Spaces in Urban Subway Stations"

_buildings, doi:10.3390/buildings12122209_

Round 1
Reviewer 1 Report
The authors aim at evaluating the transfer space environment of 8 sample subway stations. Using a fuzzy comprehensive evaluation method, the research establishes a system for evaluating the environmental suitability of the transfer space using primary indicators and 17 secondary indicators.
The work is written in plain English, cutting through complicated terminology in a way that will appeal to readers. The introduction provides a complete background, and the manuscript has a clear and well-structured design of the methodology.
I do not see major faults in the work. A part for a few flaws as follows:
- The abbreviation AHP occurs many times (lines 43, 47, 50, 52 and 149) in the manuscript without an explanation of the meaning. Authors should avoid using abbreviations without giving disambiguation
- the graphic quality of figure 1 should be improved. In particular, the resolution of the three images
- Lines 137-139 The sentence is unclear, probably a typo.
- Just a style tip. The colour scale of figure 5 should be reversed. Normally, red in colour ramps indicates something negative, while green relates to something positive.
Reviewer 2 Report
The paper by Zihan Wu, Xiang Ji, Xiaochun Hong, Xinyu Liu, Yaxi Gong presents an environmental suitability evaluation study using fuzzy comprehensive evaluation and applies it to 8 subway stations in Shanghai. Results show that safety and convenience have higher weight among the 5 first-level index weights.
The reviewer thanks the authors for the well conducted research, a few comments are given below.
1. Hernandez et al. missing reference in line 39.
2. Figure 1 is illegible, consider adding an image with better quality, expanding the figure to wide page or presenting the different images separately.
3. In section 2.2, for foreigner readers, explain that WeChat is the most commonly used messaging application in China.
4. End of section 2.2. Maybe better “information significance” instead of effectiveness?
5. Section 3.1. what do you mean by “spiritual needs”?
6. Explain equations 2 and 4.
7. Equations in section 3.3 are not numbered.
8. Figure 8 vertical axis lacks label.
9. Lines 423-425 “research conclusion” refers to the previous evaluations from bibliography or present work conclusion? Next lines mention the effect of the epidemic situation, how did authors evaluate the effect of epidemics?
10. Why is the achievement of “energy saving and sustainable development” an objective in line 460? This is not mentioned anywhere else in the document.
11. How relevant is the negative correlation with various indicators in figures 19 and 20 considering that the correlation coefficient R2 is in all cases very low? Is it worth mentioning anything about those correlations?
12. Some images have low quality, if you can export them with higher resolution or vector format they can be improved.
Reviewer 3 Report
The current paper analyzes the factors affecting the conditions of transfer space in urban subway station. The paper is well-written, discussing an interesting topic which can be helpful for decision-makers. The presentation of methodology and results are straightforward and easy to follow. I would like to ask authors to add some up-to-dated references to better show the research gap and the paper`s novelty.
Reviewer 4 Report
This is a good paper, certainly deserving a good deal of attention from this journal. I have only few suggestions for the authors:
i) I would recap some of the previous attempts dealing with transit quality, inserting the argument of the paper within this wider frame. This helps much the readers to focus on the effective contribution of the manuscript to the literature. You may add contents from: Newman, P.W.G., Kenworthy, J.R. (1996). The land use-transport connection: An overview. Land Use Policy 13(1), pp. 1-22; Nocera S. (2010). An Operational Approach for Quality Evaluation in Public Transport Services. Ingegneria Ferroviaria 65-4: 363-383; Nocera, S. (2011). The key role of quality assessment in public transport policy. Traffic Engineering & Control 52-9: 394-398; Tyrinopoulos, Y., Antoniou, C. (2008). Public transit user satisfaction: Variability and policy implications. Transport Policy 15(4), pp. 260-272;
ii) Section 2 needs to be widened much. This is the core of the paper, and the method as explained here is not clear at all. Particularly, the sentence: “In order to ensure the authenticity of the data source, a combination of various research methods such as actual measurement of transfer routes, activity observation, and questionnaire interviews were used to calculate the time of different transfer routes for each sample based on the average value of field walking of various types of people. The accuracy of the research results was verified by data comparison (lines 107-110)” needs further explanation;
iii) I feel like the choice of the indicators in section 3 needs also to be further explained. Adding some literature source can probably help this argument;
iv) In the conclusions, I would dwell a little bit on the possibility of using this method outside of the chosen stations. It must be clear, whether we are dealing with a method essentially conceived to solve the problem presented, or whether it is possible to generalize it to other similar cases.
Reviewer 5 Report
Matrix equations (Eq.1-5, etc.) contain typos, most likely the result of Word=>PDF conversion (M and Lambda instead of ellipsis, etc.)
Round 2
Reviewer 4 Report
Paper ready for publication